# Relationships between hemoglobin levels at admission and adverse maternal and perinatal outcomes in patients with preeclampsia

**Yanan Lian, Yanxiang Lv, Yuan Qiao, Tongqiang He**[iD]*

Department of Obstetrics and Gynecology Intensive Care Unit, Northwest Women's and Children's Hospital, Xi'an City, Shaanxi Prov, China

* xbfymicu@163.com

## Abstract

### Background

Maternal hemoglobin is very important for maternal and perinatal outcomes. Due to the pathophysiological changes in patients with preeclampsia, the influence of hemoglobin on pregnancy outcomes may differ from that in normal pregnant women. Therefore, this retrospective study aimed to evaluate the relationships between maternal hemoglobin levels and adverse maternal and perinatal outcomes in patients with preeclampsia.

### Methods

All clinical data were retrospectively collected from the medical records of a tertiary obstetrics and gynecology hospital in China. This study evaluated the incidence of adverse maternal and perinatal outcomes in patients with preeclampsia with different hemoglobin levels at admission. The odds ratios and 95% confidence intervals for adverse pregnancy outcomes in patients with preeclampsia with anemia and high hemoglobin levels were estimated, with the normal hemoglobin level serving as the control.

### Results

A total of 1,715 patients with preeclampsia with singleton pregnancies were included in this retrospective study. Compared with patients with preeclampsia with normal hemoglobin levels, patients with anemia at admission had a greater risk for post-partum hemorrhage (OR: 3.800; 95% CI: 1.677–8.610) and cardiac dysfunction (OR: 2.860; 95% CI: 0.979–8.356). Moreover, patients with high hemoglobin levels at admission had increased risks of HELLP syndrome (OR: 2.503; 95% CI: 1.198–5.229), SGA (OR: 1.343; 95% CI: 0.997–1.808), neonatal asphyxia (OR: 2.046; 95% CI: 1.107–3.784) and NICU admission (OR: 1.359; 95% CI: 1.060–1.742). However,

**Data availability statement:** Data cannot be shared publicly because of privacy and ethical restrictions. Data are available from the Ethics Committee of Northwest Women's and Children's Hospital (contact via xbfyyxll@163.com) for researchers who meet the criteria for access to confidential data.

**Funding:** This project is supported by the Key Research and Development Program of Shaanxi China (Program number 2024SF-YBXM-230). The funders have no role in study design, data collection and analysis, the decision to publish, or the preparation of the manuscript. No author receives a salary from any of the funders.

**Competing interests:** The authors have declared that no competing interests exist.

not all abnormal hemoglobin levels were associated with an increased risk of adverse pregnancy outcomes. Patients with preeclampsia with anemia had a lower risk of adverse perinatal outcomes, including SGA (OR: 0.731; 95% CI: 0.517–1.032) and NICU admission (OR: 0.737; 95% CI: 0.567–0.960).

## Conclusion

This study revealed that both anemia and high hemoglobin levels at admission were related to adverse maternal and perinatal outcomes in patients with preeclampsia. The effects of hemoglobin on adverse maternal and perinatal outcomes in patients with preeclampsia may differ from those in normal pregnant women.

---

## Introduction

Maternal hemoglobin is very important for both pregnant women and perinatal infants [1–3]. The World Health Organization defines anemia during pregnancy as a hemoglobin level below 110 g/L. Anemia during pregnancy is a prevalent global health concern, with a worldwide prevalence of approximately 40%. A nationwide cross-sectional study conducted in China revealed that 19.8% of pregnant women were diagnosed with anemia [4]. Several studies have demonstrated an association between anemia and a series of adverse pregnancy complications, such as postpartum hemorrhage, preterm birth, low birth weight of newborns, and infections [5,6]. However, high hemoglobin levels in pregnant patients do not invariably signify an optimistic prognosis. Elevated hemoglobin levels have been demonstrated to be associated with a variety of adverse maternal and perinatal outcomes, including stillbirth, preeclampsia, gestational diabetes and small for gestational age (SGA) [7–9]. Preeclampsia is a pregnancy complication characterized by hypertension and proteinuria/organ damage. In addition, preeclampsia is one of the main reasons for maternal and neonatal morbidity and mortality, with an incidence rate of approximately 3%–5% of all pregnancies [10,11]. The pathophysiological changes associated with preeclampsia include vasospasm, endothelial dysfunction, and vascular leakage, which may lead to clinical manifestations such as plasma volume depletion, elevated hemoglobin level, and hypertension. These changes can result in impaired uteroplacental blood flow, exacerbated inflammatory responses, and ultimately adverse pregnancy outcomes [12–14]. Hemoglobin levels in patients with preeclampsia are not only related to their nutritional status and iron intake during pregnancy but also related to the pathophysiological changes [15].

Therefore, the effects of hemoglobin levels on pregnancy outcomes may differ between pregnant women with preeclampsia and normal pregnant women. However, data regarding the associations of maternal and perinatal complications with different hemoglobin levels in patients with preeclampsia are limited. Moreover, extant studies have focused primarily on the impact of anemia on pregnancy outcomes, while relatively little research has been conducted to the impact of elevated hemoglobin levels on pregnancy outcomes [9]. Hence, we conducted a retrospective study to evaluate

the relationships between different maternal hemoglobin levels measured at admission and adverse maternal and perinatal outcomes in patients with preeclampsia.

## Methods

### Ethics approval

This study complied with the Declaration of Helsinki and was approved by the Ethics Committee of Northwest Women's and Children's Hospital (approval number: 2024−012). Informed consent was waived by the ethics committee due to the retrospective nature of the study. All patient information was anonymized during the analysis.

### Study design

All clinical data of patients with preeclampsia admitted to the Obstetrics and Gynecology Intensive Care Unit of Northwest Women's and Children's Hospital between 01/01/2018 and 31/12/2023 were retrospectively collected from medical records. The clinical data for this study were collected between 01/03/2024 and 31/05/2024 after approval by the ethics committee. The Northwest Women's and Children's Hospital is a tertiary obstetrics and gynecology hospital and one of the main maternal care referral hospitals in Shaanxi Province. Over the past seven years, our hospital has delivered more than 20,000 infants annually. For each patient, maternal hemoglobin level data at admission were obtained. Clinical characteristics of patients with preeclampsia, including maternal age, pre-pregnancy body mass index (BMI), gestational age at the onset of preeclampsia, and the presence or absence of the following: severe preeclampsia, nulliparas, and assisted reproduction, were obtained. The gestational age at the termination of pregnancy and the birth weight of the newborns were also recorded.

In this study, we defined gestational anemia as a hemoglobin level ≤ 109 g/L, high hemoglobin level as a hemoglobin level ≥ 130 g/L, and normal hemoglobin level as a hemoglobin level between 110 g/L and 129 g/L [9,16]. Hemoglobin was measured at the laboratory department of Northwest Women's and Children's Hospital, employing the BC-7500CRP of Mindray (China). Preeclampsia was defined as new-onset hypertension (systolic blood pressure ≥ 140 and/or diastolic blood pressure ≥ 90 mm Hg) with proteinuria (24-h urine protein exceeding 300 mg or urine routine protein level of 1+ or greater) or other maternal organ dysfunction after 20 weeks of gestation, including abnormal liver function (alanine aminotransferase or aspartate aminotransferase > 40 IU/L), abnormal renal function tests (creatinine ≥ 90 μmol/L), abnormal neurological complications (eclampsia, blindness, severe headaches, and persistent visual scotomata, etc.), thrombocytopenia, fetal growth restriction, oligohydramnios, stillbirth, etc. [17]. In addition, the exclusion criteria included preexisting chronic hypertension, multiple pregnancies, an absence of hemoglobin data at admission, and missing pregnancy outcome data. The specific exclusion process is presented in Fig 1. The patients with preeclampsia included in the study were divided into three groups on the basis of their hemoglobin levels at admission: Low Hb group, Normal Hb group and High Hb group.

### Study outcomes

The adverse maternal outcomes that were evaluated in this study included cardiac dysfunction, eclampsia, postpartum hemorrhage, cesarean delivery, placental abruption and hemolysis, elevated liver enzymes, and thrombocytopenia syndrome (HELLP syndrome).

The adverse perinatal outcomes included in this study were as follows: intrauterine fetal death (IUFD, defined as fetal death occurring in the uterus after 20 weeks of gestation), SGA (defined as a birth weight below the 10th percentile for newborns of the same gestational age), premature delivery (delivery before 37 weeks of gestation), neonatal asphyxia (defined as an Apgar score ≤ 7 at 5 minutes post-delivery), and admission to the neonatal intensive care unit (NICU).

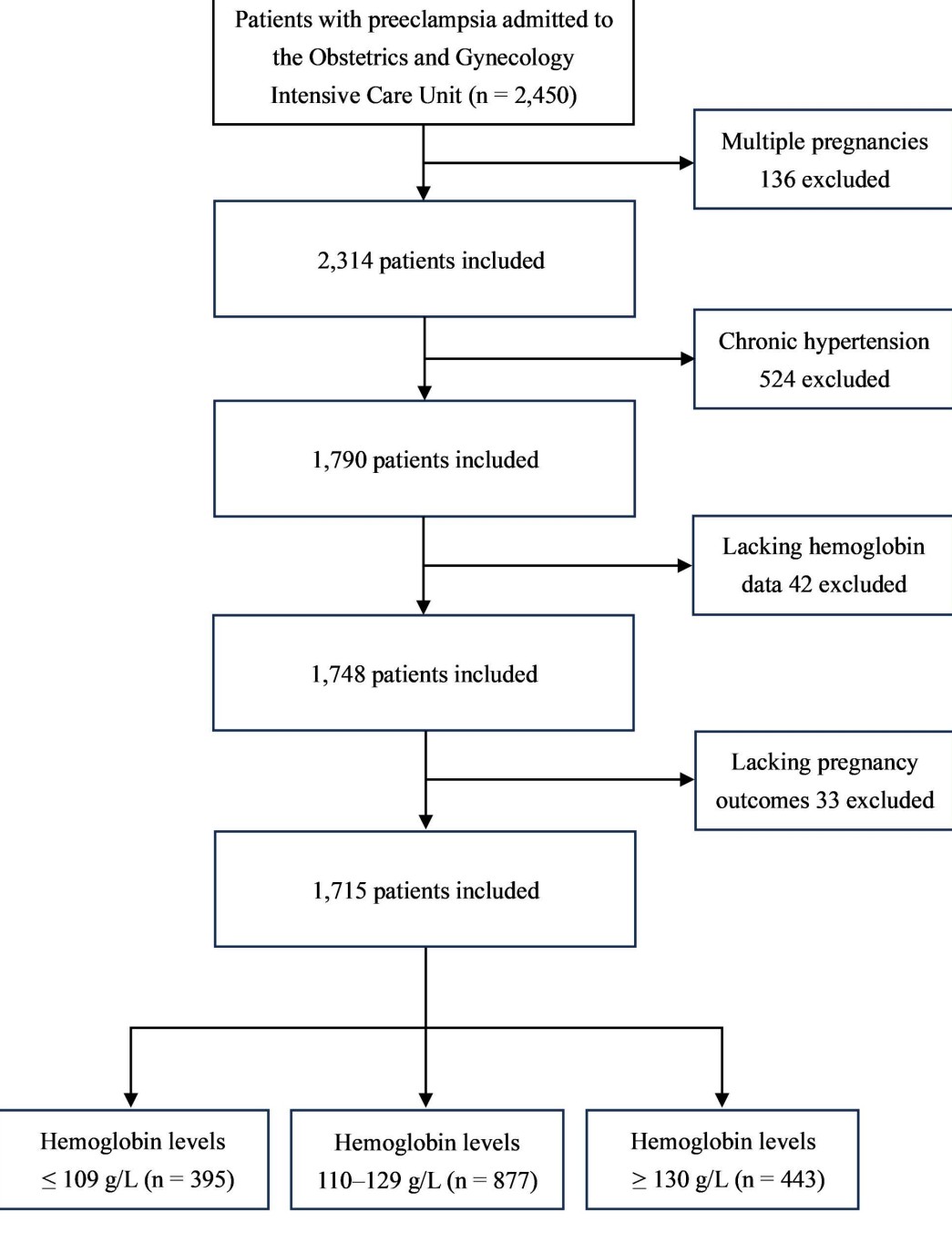

**Fig 1. Flow chart of the study population.**

## Statistical analysis

Continuous variables are expressed as means and standard deviations (SDs). The differences in continuous variables among the three groups were compared using analysis of variance. Categorical variables are expressed as frequencies

and percentages. The differences in categorical data were analyzed for statistical significance using the chi-square test or Fisher's exact test.

We separately calculated the incidences and 95% confidence intervals (CIs) of adverse maternal and perinatal outcomes in patients with preeclampsia with anemia, normal hemoglobin levels, and high hemoglobin levels. The differences in the incidences of adverse pregnancy outcomes among the three groups were compared using the chi-square test. We performed a post hoc power analysis to examine the effects of different hemoglobin levels on adverse pregnancy outcomes using R language (version 4.4.2). Using hemoglobin levels of 110–129 g/L as the reference category, we estimated odds ratios (ORs) with 95% CIs for complications within the anemia and high hemoglobin level categories using binary logistic regression in SPSS. To minimize the effects of confounding factors on the effects of anemia and high hemoglobin levels on adverse pregnancy outcomes in patients with preeclampsia, we included maternal age, pre-pregnancy BMI, gestational age at the onset of preeclampsia and the presence of severe preeclampsia, nulliparas, and assisted reproduction as covariates in our binary logistic regression analysis. A two-sided P < 0.05 was considered to indicate statistical significance. All the statistical analyses were performed using the IBM SPSS, Inc. (Chicago, USA) statistical software package, version 27.0.

## Results

### Baseline characteristics of the study population

A total of 2,450 patients with preeclampsia were included in this study. Among them, 136 cases were excluded because of multiple pregnancies, 524 cases were excluded because of chronic hypertension superimposed on preeclampsia, 42 cases were excluded because of a lack of hemoglobin data at admission, and a further 33 cases were excluded due to missing pregnancy outcome data. (Fig 1). Ultimately, 1,715 patients with preeclampsia with singleton pregnancies were included in the study. The mean hemoglobin level at admission was 120.27 g/L (SD 15.52) for all patients included in the study, and the mean gestational age at the onset of preeclampsia was 33.98 weeks (SD 4.18).

The study population comprised 395 patients in the Low Hb group, 877 in the Normal Hb group, and 443 in the High Hb group. The clinical characteristics of patients with preeclampsia with different hemoglobin levels are shown in Table 1. The clinical data in this study revealed no significant differences in age, pre-pregnancy BMI, gestational age at onset of preeclampsia, gestational age at termination of pregnancy, and the proportion of assisted reproductive pregnancies and nulliparas among the three groups of patients with preeclampsia (all P > 0.05). However, among patients with preeclampsia, the proportion of severe preeclampsia was greater in the High Hb group than in the Low Hb group and the Normal Hb group. Furthermore, despite the absence of discrepancies in gestational age at the termination of pregnancy among

**Table 1. Baseline clinical characteristics of the study population.**

| Characteristics | Low Hb group (n = 395) | Normal Hb group (n = 877) | High Hb group (n = 443) | Statistics | P value |
|---|---|---|---|---|---|
| Age, years | 31.17 ± 4.63 | 31.41 ± 4.43 | 31.38 ± 4.18 | $F = 0.419$ | 0.658 |
| Pre-pregnancy BMI, kg/m² | 23.84 ± 4.35 | 24.01 ± 4.24 | 23.60 ± 3.84 | $F = 1.450$ | 0.235 |
| Gestational age at the onset of preeclampsia, weeks | 33.89 ± 4.24 | 33.97 ± 4.23 | 34.08 ± 4.02 | $F = 0.234$ | 0.791 |
| Severe preeclampsia, n (%) | 359 (90.9) | 781 (89.1) | 414 (93.5) | $\chi^2 = 6.744$ | 0.034 |
| Nulliparas, n (%) | 222 (56.2) | 534 (60.9) | 275 (62.1) | $\chi^2 = 3.452$ | 0.178 |
| Assisted reproduction, n (%) | 37 (9.4) | 103 (11.7) | 59 (13.3) | $\chi^2 = 3.213$ | 0.201 |
| Gestational age at termination of pregnancy, weeks | 34.98 ± 3.64 | 35.01 ± 3.60 | 34.88 ± 3.55 | $F = 0.184$ | 0.832 |
| Birth weight of newborns, g | 2317.78 ± 918.94 | 2267.58 ± 878.51 | 2139.80 ± 868.42 | $F = 4.746$ | 0.009 |

BMI, body mass index.

the three groups examined in this study, an inverse correlation was observed between the birth weight of newborns and increasing hemoglobin levels.

### Incidence of adverse pregnancy outcomes in patients with preeclampsia with different hemoglobin levels

We compared the incidence of adverse maternal and perinatal outcomes among patients with preeclampsia with three different hemoglobin levels. Among patients with preeclampsia, there were significant differences in postpartum hemorrhage and NICU admission rates among the three groups, and the post hoc power was sufficient. Statistically significant differences in adverse maternal outcomes, such as eclampsia, and HELLP syndrome, as well as adverse perinatal outcomes, such as IUFD and SGA, were detected among patients with preeclampsia with different hemoglobin levels, although the post hoc power was suboptimal. These results are presented in Table 2.

Using normal hemoglobin levels as a reference, we calculated ORs and 95% CIs for the risk of maternal and perinatal complications for both anemia and high hemoglobin levels (Fig 2). After adjusting for confounding factors such as maternal age, pre-pregnancy BMI, gestational age at the onset of preeclampsia, severe preeclampsia, nulliparas and assisted reproduction, the results of the multivariate regression analysis indicated that certain adverse pregnancy outcomes were associated with anemia or high hemoglobin levels. Compared with patients with preeclampsia with normal hemoglobin levels, patients with anemia at admission were at a greater risk for postpartum hemorrhage (OR: 3.800; 95% CI: 1.677–8.610) and cardiac dysfunction (OR: 2.860; 95% CI: 0.979–8.356). Moreover, patients with high hemoglobin levels at admission were at an increased risk for developing HELLP syndrome (OR: 2.503; 95% CI: 1.198–5.229). With respect to perinatal outcomes, we found that patients with preeclampsia with high hemoglobin levels were at an increased risk of SGA and NICU admission, whereas patients with anemia were at a reduced risk. In addition, patients with preeclampsia with high hemoglobin levels were at an increased risk of neonatal asphyxia (OR: 2.046; 95% CI: 1.107–3.784).

## Discussion

### Main findings

Anemia and elevated hemoglobin levels are associated with a variety of adverse maternal and perinatal outcomes [9,18]. Consistent with previous studies in normal pregnant women, this study revealed that both anemia and high hemoglobin

**Table 2. Incidence of adverse maternal and perinatal outcomes in patients with preeclampsia with three hemoglobin levels.**

| Variables | Low Hb group (n=395) | | Normal Hb group (n=877) | | High Hb group (n=443) | | $\chi^2$ | P | Post hoc power, % |
|---|---|---|---|---|---|---|---|---|---|
| | n | Rate, % (95% CI) | n | Rate, % (95% CI) | n | Rate, % (95% CI) | | | |
| Maternal | | | | | | | | | |
| Cardiac dysfunction | 8 | 2.0 (0.9, 4.0) | 6 | 0.7 (0.3, 1.5) | 8 | 1.8 (0.8, 3.5) | 5.159 | 0.076 | 52.4 |
| Eclampsia | 2 | 0.5 (0.1, 1.8) | 9 | 1.0 (0.5, 1.9) | 11 | 2.5 (1.2, 4.4) | 7.376 | 0.025 | 67.6 |
| Postpartum hemorrhage | 15 | 3.8 (2.1, 6.2) | 11 | 1.3 (0.6, 2.2) | 4 | 0.9 (0.2, 2.3) | 12.738 | 0.002 | 90.1 |
| HELLP syndrome | 9 | 2.3 (1.0, 4.3) | 13 | 1.5 (0.8, 2.5) | 17 | 3.8 (2.3, 6.1) | 7.346 | 0.025 | 67.5 |
| Placental abruption | 23 | 5.8 (3.7, 8.6) | 51 | 5.8 (4.4, 7.6) | 23 | 5.2 (3.3, 7.7) | 0.241 | 0.886 | 6.8 |
| Cesarean delivery | 362 | 91.6 (88.5, 94.2) | 779 | 88.8 (86.6, 90.8) | 401 | 90.5 (87.4, 93.1) | 2.63 | 0.268 | 28.7 |
| Perinatal | | | | | | | | | |
| Premature delivery | 229 | 58 (52.9, 62.9) | 506 | 57.7 (54.3, 61.0) | 278 | 62.8 (58.1, 67.3) | 3.367 | 0.186 | 35.7 |
| IUFD | 15 | 3.8 (2.1, 6.2) | 14 | 1.6 (0.9, 2.7) | 6 | 1.4 (0.5, 2.9) | 8.008 | 0.018 | 70.9 |
| SGA | 56 | 14.2 (10.9, 18.0) | 154 | 17.6 (15.1, 20.2) | 97 | 21.9 (18.1, 26.0) | 8.608 | 0.014 | 75.1 |
| Neonatal asphyxia | 14 | 3.5 (2.0, 5.9) | 24 | 2.7 (1.8, 4.0) | 23 | 5.2 (3.3, 7.7) | 5.173 | 0.075 | 51.6 |
| NICU admission | 158 | 40 (35.1, 45.0) | 396 | 45.2 (41.8, 48.5) | 231 | 52.1 (47.4, 56.9) | 12.684 | 0.002 | 90.1 |

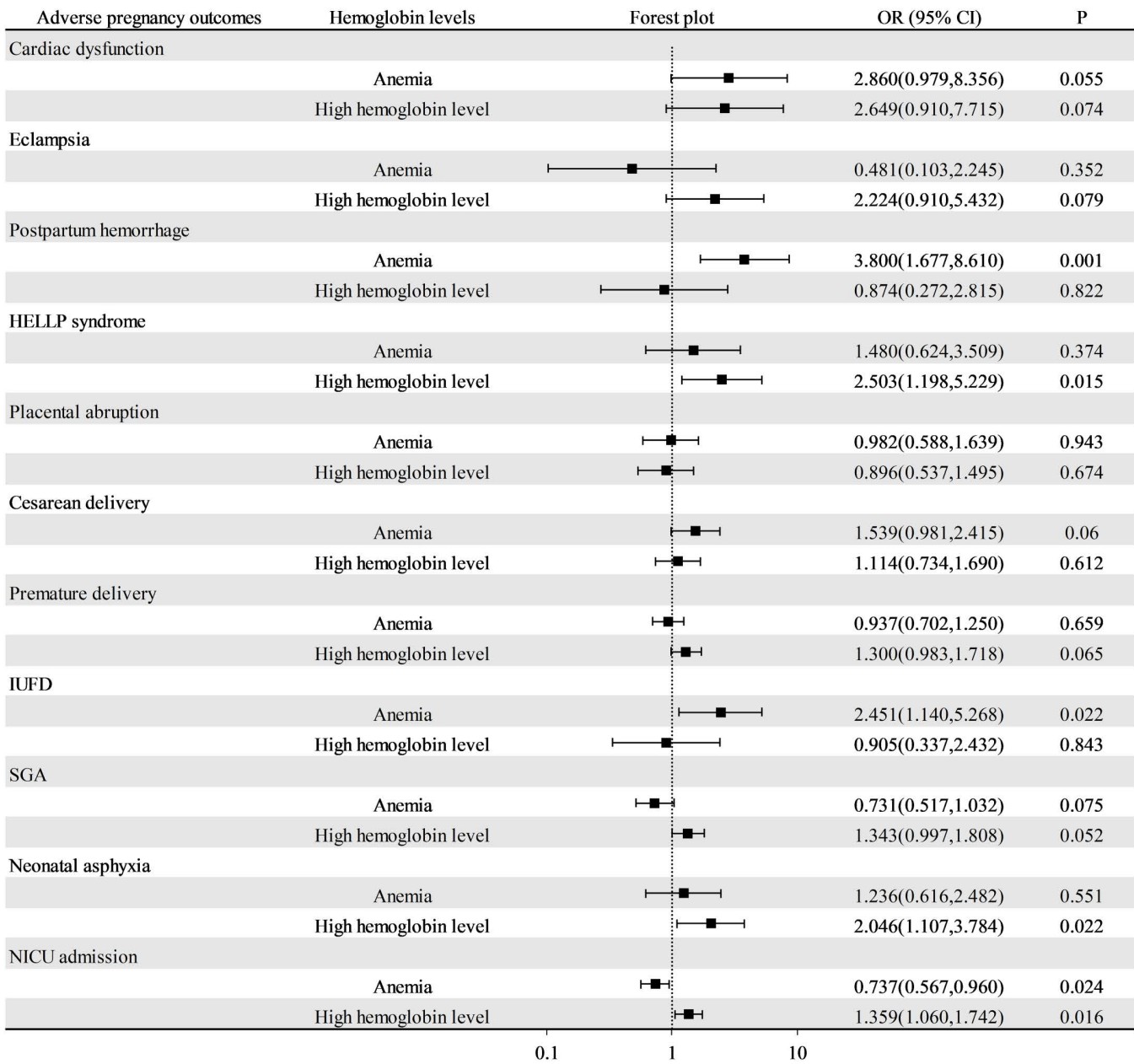

| Adverse pregnancy outcomes | Hemoglobin levels | Forest plot | OR (95% CI) | P |
|---|---|---|---|---|
| **Cardiac dysfunction** | | | | |
| | Anemia | | 2.860(0.979,8.356) | 0.055 |
| | High hemoglobin level | | 2.649(0.910,7.715) | 0.074 |
| **Eclampsia** | | | | |
| | Anemia | | 0.481(0.103,2.245) | 0.352 |
| | High hemoglobin level | | 2.224(0.910,5.432) | 0.079 |
| **Postpartum hemorrhage** | | | | |
| | Anemia | | 3.800(1.677,8.610) | 0.001 |
| | High hemoglobin level | | 0.874(0.272,2.815) | 0.822 |
| **HELLP syndrome** | | | | |
| | Anemia | | 1.480(0.624,3.509) | 0.374 |
| | High hemoglobin level | | 2.503(1.198,5.229) | 0.015 |
| **Placental abruption** | | | | |
| | Anemia | | 0.982(0.588,1.639) | 0.943 |
| | High hemoglobin level | | 0.896(0.537,1.495) | 0.674 |
| **Cesarean delivery** | | | | |
| | Anemia | | 1.539(0.981,2.415) | 0.06 |
| | High hemoglobin level | | 1.114(0.734,1.690) | 0.612 |
| **Premature delivery** | | | | |
| | Anemia | | 0.937(0.702,1.250) | 0.659 |
| | High hemoglobin level | | 1.300(0.983,1.718) | 0.065 |
| **IUFD** | | | | |
| | Anemia | | 2.451(1.140,5.268) | 0.022 |
| | High hemoglobin level | | 0.905(0.337,2.432) | 0.843 |
| **SGA** | | | | |
| | Anemia | | 0.731(0.517,1.032) | 0.075 |
| | High hemoglobin level | | 1.343(0.997,1.808) | 0.052 |
| **Neonatal asphyxia** | | | | |
| | Anemia | | 1.236(0.616,2.482) | 0.551 |
| | High hemoglobin level | | 2.046(1.107,3.784) | 0.022 |
| **NICU admission** | | | | |
| | Anemia | | 0.737(0.567,0.960) | 0.024 |
| | High hemoglobin level | | 1.359(1.060,1.742) | 0.016 |

**Fig 2. Odds ratios for adverse pregnancy outcomes associated with anemia and high hemoglobin levels.** OR, odds ratio; CI, confidence interval; HELLP syndrome, hemolysis, elevated liver enzymes, and thrombocytopenia syndrome; IUFD, intrauterine fetal death; SGA, small for gestational age.

levels were associated with certain adverse maternal and perinatal outcomes in patients with preeclampsia. Notably, our study revealed that anemia increases the risk of adverse maternal complications, such as cardiac dysfunction and postpartum hemorrhage, in patients with preeclampsia. However, anemia is a protective factor against certain perinatal outcomes (including SGA and NICU admission). Although not all the differences were statistically significant, our study revealed an association between high hemoglobin levels and adverse maternal and perinatal outcomes in patients with preeclampsia. These complications included cardiac dysfunction, eclampsia, HELLP syndrome, SGA, neonatal asphyxia,

and NICU admission. Our findings suggest that for patients with preeclampsia, high hemoglobin levels have a broader and more serious effect on adverse pregnancy outcomes, particularly adverse perinatal outcomes. This phenomenon may be caused by specific pathophysiological changes in patients with preeclampsia. It is well known that the increase in plasma volume exceeds that of red blood cell volume during pregnancy, a critical factor in maintaining fetal growth and development. This physiological change in blood volume may lead to physiological anemia, which in turn helps maintain a sustained increase in uterine and placental blood flow [19]. In patients with preeclampsia, lower hemoglobin levels are more stable and are associated with reduced hemodynamic changes [20]. The fundamental pathophysiological changes in preeclampsia are vasospasm, endothelial dysfunction, and vascular leakage. Elevated hemoglobin levels in patients with preeclampsia may lead to increased blood viscosity and reduced uterine and placental blood flow. These changes can exacerbate inflammatory responses and immune dysregulation, thereby contributing to adverse pregnancy outcomes [12–14]. Therefore, monitoring hemoglobin levels in patients with preeclampsia is highly clinically important. Obstetricians can determine clinical management priorities and indicators requiring close monitoring on the basis of patients' hemoglobin levels and the severity of their preeclampsia. This will help reduce the risk of negative maternal and perinatal outcomes.

### Hemoglobin levels and adverse maternal outcomes

Anemia increases the incidence of postpartum hemorrhage, and the incidence increases with the severity of anemia [21–23]. Similar results were observed in patients with preeclampsia in our study, where anemia was associated with an elevated risk of postpartum hemorrhage (OR: 3.800; 95% CI: 1.677–8.610). In our study, the relationship between anemia and postpartum hemorrhage in patients with preeclampsia showed sufficient post hoc power in the initial chi-square analysis and remained statistically significant in the subsequent multivariate regression analysis. This finding confirmed the reliability of the results. However, due to limitations in sample size, we could not analyze the impact of varying degrees of anemia on postpartum hemorrhage in patients with preeclampsia. Further multicenter, large-scale research is necessary to clarify the specific effects of different degrees of anemia on postpartum hemorrhage in patients with preeclampsia.

Compared with those of healthy pregnant women, patients with preeclampsia have lower ejection fractions and cardiac output and higher peripheral vascular resistance [24,25], resulting in a significantly increased risk of cardiac dysfunction in patients with preeclampsia. However, the mechanisms underlying cardiac dysfunction in patients with preeclampsia remain unclear. The proposed mechanisms include hypertension, oxidative stress, inflammation, persistent endothelial dysfunction, and coronary microvascular dysfunction [26]. Previous studies have suggested that people with anemia are at a significantly greater risk of developing cardiac dysfunction than those without anemia [27]. Consistent with prior studies, we found that anemia in patients with preeclampsia is related to an elevated risk of cardiac dysfunction. These findings indicate that anemia may contribute to cardiac dysfunction in patients with preeclampsia, with potential mechanisms including a reduced oxygen-carrying capacity of the blood, leading to compensatory cardiac overload, while the myocardium remains in a relatively hypoxic state. Additionally, anemia during pregnancy may induce oxidative stress and reduce the left ventricular ejection fraction [28]. Interestingly, patients with preeclampsia with high hemoglobin levels also face a greater risk of cardiac dysfunction. However, this association was not statistically significant according to multivariate regression analysis and may be related to the more severe condition of patients with preeclampsia with high hemoglobin levels.

Eclampsia is a serious complication of pregnancy, with elevated maternal mortality and morbidity rates, potentially resulting in placental abruption, HELLP syndrome, disseminated intravascular coagulopathy, cardiopulmonary arrest, and other complications [29]. Our study revealed that patients with preeclampsia with high hemoglobin levels had a greater incidence of eclampsia than those with normal hemoglobin levels or anemia did. After adjustment for confounding factors (including maternal age, gestational age at onset, severity of preeclampsia, and other relevant factors), a high hemoglobin level was related to an elevated risk of eclampsia compared with a normal hemoglobin level (OR: 2.224; 95% CI:

0.910–5.432). This observation may be attributed to the fact that patients with preeclampsia with high hemoglobin levels tend to have more severe conditions. However, Liu S et al. reported that anemia is a contributing factor to eclampsia [30]. In our study, patients with preeclampsia who had anemia had a lower risk of developing eclampsia than those with high hemoglobin levels did. This discrepancy may be attributable to the inclusion of different study populations. Our study focused on pregnant women who had already developed preeclampsia, in contrast to the broader population of pregnant individuals examined in the aforementioned study. Further studies are needed to explore the role of hemoglobin in the development of eclampsia in patients with preeclampsia.

HELLP syndrome is a severe complication of pregnancy that can result in maternal and fetal morbidity and mortality [31,32]. The exact pathogenesis of HELLP syndrome, which involves genetic factors, inflammatory factors, immune factors, abnormal oxidative stress, and abnormal lipid metabolism, remains unclear. In this study, high hemoglobin levels in patients with preeclampsia were related to an elevated risk of HELLP syndrome (OR: 2.503; 95% CI: 1.198–5.229). These findings contrast with the results reported by Huang et al., who reported that patients with HELLP syndrome had lower hemoglobin levels than those without HELLP syndrome did [33]. This discrepancy may be attributable to variations in the timing of blood tests. The hemoglobin data presented herein were collected prior to the onset of HELLP syndrome, while their data were not. Hemolysis can result in a substantial reduction in hemoglobin levels following the onset of HELLP syndrome. The underlying mechanism that generates this association remains to be elucidated. Further research is necessary to elucidate the mechanism by which high hemoglobin levels increase the risk of HELLP syndrome in patients with preeclampsia.

### Hemoglobin levels and adverse perinatal outcomes

The presence of high hemoglobin levels and anemia have been demonstrated to be related to various negative perinatal complications, such as stillbirth, SGA, and low birth weight of newborns [9,34]. Our study revealed that high hemoglobin levels in patients with preeclampsia were associated with SGA, neonatal asphyxia, and NICU admission. Furthermore, an inverse correlation was identified between the birth weight of newborns and increasing hemoglobin levels in patients with preeclampsia, which is consistent with the findings of previous literature reports [35]. This phenomenon may be related to increased blood viscosity caused by high hemoglobin levels, which in turn leads to inadequate uteroplacental blood perfusion. Interestingly, although anemia may be a protective factor against SGA and NICU admission in women with preeclampsia, it concomitantly increased the risk of IUFD (OR: 2.451; 95% CI: 1.140–5.268). This may be because anemia in patients with preeclampsia leads to decreased blood viscosity, which can improve uteroplacental blood flow. In cases where anemia is not severe, the overall oxygen supply to the uterus and placenta may be better than that in patients with preeclampsia with elevated hemoglobin levels. However, as the hemoglobin level continues to decrease, the ability of the blood to carry oxygen decreases as well. Ultimately, this leads to a reduced oxygen supply to the uterus and placenta, increasing the risk of IUFD [36,37]. These results indicate that for patients with preeclampsia with high hemoglobin levels, greater attention should be given to their perinatal status, and for patients with preeclampsia with anemia, vigilance is needed regarding the risk of IUFD. However, our sample size was insufficient to further explore the optimal hemoglobin level for perinatal outcomes in patients with preeclampsia. Further multicenter studies with larger sample sizes are needed to clarify the optimal hemoglobin level for different adverse perinatal outcomes.

### Strengths and limitations

The primary strength of this study was its utilization of normal hemoglobin levels as a reference standard for the analysis of the effects of anemia and high hemoglobin levels on adverse pregnancy outcomes in patients with preeclampsia. However, this research has several limitations. First, we were unable to analyze the long-term influence of hemoglobin levels on mothers and newborns because of the absence of follow-up data. Second, as we are a referral obstetrics hospital in

Shaanxi Province, we receive a large volume of critically ill pregnant women who have been transferred from primary care hospitals, resulting in an excessively high proportion of severe preeclampsia cases in the study data. Therefore, the conclusions of this study may be more applicable to patients with severe preeclampsia. Third, due to the low incidence of certain adverse pregnancy outcomes, the post hoc power of some results is insufficient. Although we included major confounding factors in the binary logistic regression, there may still be potential covariates influencing the results that were not included in the analysis. The observed associations require further confirmation through larger-scale and multicenter studies. Finally, due to the insufficient sample size, the categorization of hemoglobin level was limited to three groups, thereby restricting the investigation of optimal hemoglobin levels in patients with preeclampsia.

## Conclusion

Our study emphasized the importance of maternal hemoglobin levels for pregnancy outcomes in patients with preeclampsia. In this retrospective study, we found that both anemia and high hemoglobin levels were related to adverse maternal and perinatal complications. Our research emphasized that in addition to monitoring the effect of anemia on pregnancy outcomes, we should also be aware of the negative influence of high hemoglobin levels on pregnancy outcomes in patients with preeclampsia. However, the mechanisms underlying these associations remain unclear. It is challenging to determine whether the negative pregnancy complications identified in patients with preeclampsia are attributable to hemoglobin levels or the severity of preeclampsia itself. Additional research is necessary to validate the optimal hemoglobin levels and elucidate the role of hemoglobin levels in the occurrence of adverse maternal and perinatal outcomes in patients with preeclampsia.

## Acknowledgments

We are grateful to all the authors of the study.

## Author contributions

**Conceptualization:** Yanan Lian, Tongqiang He.

**Data curation:** Yanxiang Lv, Yuan Qiao.

**Formal analysis:** Yanan Lian.

**Funding acquisition:** Tongqiang He.

**Investigation:** Yanan Lian, Yanxiang Lv, Yuan Qiao.

**Methodology:** Yanan Lian, Tongqiang He.

**Project administration:** Tongqiang He.

**Writing – original draft:** Yanan Lian.

**Writing – review & editing:** Yanan Lian, Tongqiang He.

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
