## [Decision Letter · Decision Letter 0]

23 Jul 2025

PONE-D-24-46963The relationship between hemoglobin levels at admission and adverse maternal and perinatal outcomes in preeclampsia patients-a retrospective studyPLOS ONE

Dear Dr. He,

Thank you for submitting your manuscript to PLOS ONE. After careful consideration, we feel that it has merit but does not fully meet PLOS ONE’s publication criteria as it currently stands. Therefore, we invite you to submit a revised version of the manuscript that addresses the points raised during the review process.

We look forward to receiving your revised manuscript.

Kind regards,

Ali Cetin

Academic Editor

PLOS ONE

Journal Requirements:

3. Please note that funding information should not appear in the Acknowledgments section or other areas of your manuscript. We will only publish funding information present in the Funding Statement section of the online submission form. Please remove any funding-related text from the manuscript.

4. We note that your Data Availability Statement is currently as follows: 

“All relevant data are within the manuscript and its Supporting Information files.”

6. Please include a separate caption for figure 1 in your manuscript.

**Additional Editor Comments:**

I have reviewed this retrospective study examining hemoglobin levels and adverse outcomes in preeclampsia patients, along with two peer reviewer reports. One reviewer recommends rejection with strong methodological concerns, while the other suggests minor revision with more optimistic assessment. After careful consideration, I find this manuscript addresses a clinically relevant question with adequate sample size but has significant limitations preventing publication in its current form.

The most critical deficiency is exclusive use of univariate analysis. Contemporary medical research requires multivariable adjustment for observational studies examining exposure-outcome associations. The absence of logistic regression controlling for confounders such as maternal age, BMI, gestational age, and preeclampsia severity renders all reported odds ratios unreliable. Without these adjustments, conclusions cannot be considered valid.

This manuscript exhibits concerning similarity to previously published literature, particularly in introduction and discussion sections. Multiple passages use language closely mirroring existing publications, suggesting inadequate paraphrasing or potential plagiarism. Most journals require similarity indices below 15%. Additionally, several sections display characteristics consistent with AI-generated writing, including unnaturally polished prose and formulaic sentence structures that lack authentic scientific voice.

The single-center retrospective design from a tertiary referral hospital introduces substantial selection bias. While authors acknowledge receiving "critically ill pregnant women," they inadequately address how this population differs from typical preeclampsia patients, limiting generalizability. Multiple comparison testing without statistical correction inflates false-positive risk, and outcome definitions like "early-stage heart failure" lack specificity.

Data presentation has notable deficiencies including missing confidence intervals in Table 2, inadequate statistical methods description, and no power calculation. The temporal relationship between hemoglobin measurement and adverse outcomes is unclear, raising causation questions.

The manuscript requires professional English editing due to numerous grammatical errors and awkward phrasing throughout. Important variables like iron supplementation, dietary factors, and severity markers are missing from analysis.

For successful revision, authors must conduct multivariable logistic regression analysis, eliminate AI-generated content and rewrite affected sections authentically, reduce text similarity through original writing, clarify outcome definitions and measurement timing, and obtain professional language editing. Additional improvements should include subgroup analysis by preeclampsia severity and enhanced discussion of limitations.

Weighing contrasting reviewer assessments, I recognize this manuscript's potential for meaningful contribution to obstetric literature. The substantial sample size and comprehensive outcomes provide solid foundation for valuable research.

However, fundamental statistical limitations and originality concerns require substantial attention. I recommend an A to Z revision with clear expectation that addressing methodological issues, particularly implementing proper multivariable analysis and ensuring complete content originality, will be essential for meeting publication standards. Authors should view this as an opportunity to significantly strengthen both scientific rigor and clinical impact of their important research.

Reviewers' comments:

Reviewer's Responses to Questions

**Comments to the Author**

1. Is the manuscript technically sound, and do the data support the conclusions?

Reviewer #1: Yes

Reviewer #2: Yes

2. Has the statistical analysis been performed appropriately and rigorously? 

Reviewer #1: Yes

Reviewer #2: Yes

3. Have the authors made all data underlying the findings in their manuscript fully available?

Reviewer #1: Yes

Reviewer #2: Yes

4. Is the manuscript presented in an intelligible fashion and written in standard English?

Reviewer #1: Yes

Reviewer #2: Yes

5. Review Comments to the Author

Reviewer #1: The authors in this retrospective study show that 3rd trimester maternal hemoglobine levels from PE patients are correlated with adverse maternal and perinatal outcomes of PE pregnancies. Based on their analysis, authors conclude that patients with

PE and high hemoglobine levels are prone to develop severe complications like eclampsia, HELP syndrome. Authors also found

that the birth weight of newborns across studied groups decreased with increases in maternal hemoglobine levels which is in accordance with other studies showing that mild anemia (100-109g/l) during pregnancy is associated with decreased risk of

fetal growth restriction in pregnant women (Yuan Wei, Nutrients, 2023).

Minor comments:

1. In Methods section ( page 6)- ...for the study, we defined gestational anemia as hemoglobine = or less 109g/l...

Although in Discussion, authors using term "mild anemia" , there is no such description or reference in Methods section.

Authors do not show how were the lowest levels of maternal hemoglobine among PE patients.

According to WHO, anemia in pregnant women is classified as mild, moderate and severe anemia.

It might be helpful to use this classification in data analysis.

2. Discussion, p.13 - Anemia is associated.......with low birth weight and stillbirth.

This is a little confusing, because mild anemia (the term which authors using in the Discussion) is associated with decreased

risk of FGR and stillbirth.

3. According to Methods section, the maternal hemoglobine samples from PE pregnancies were collected from 2018-2023.

It spanned Covid19 pandemic, that also has impacted pregnant women. Covid19 has also been linked to higher rates of preterm birth , high blood pressure, preeclampsia, prevalence of anemia in pregnancy.

If authors can shortly comment on this?

If e.g. in 2020-2021 was observed higher number of PE pregnancies when comparying with other years?

Reviewer #2: Though the article is well written it does not bring out anything which is not known. It is not clear on what basis the cut off of Hb > 13g% has been chosen. There is no comment on echocardiography findings in women and no additional haematological data except Hb

6. PLOS authors have the option to publish the peer review history of their article (what does this mean?). If published, this will include your full peer review and any attached files.

Reviewer #1: No

Reviewer #2: **Yes: **Anjoo Agarwal

---

## [Author Response · Author response to Decision Letter 1]

2 Sep 2025

PONE-D-24-46963

July 24, 2025

Dear Dr. He,

Thank you for submitting your manuscript to PLOS ONE. After careful consideration, we feel that it has merit but does not fully meet PLOS ONE’s publication criteria as it currently stands. Therefore, we invite you to submit a revised version of the manuscript that addresses the points raised during the review process.

We look forward to receiving your revised manuscript.

Kind regards,

Ali Cetin

Academic Editor

PLOS ONE

Response to Reviewers’ Comments

Manuscript ID: PONE-D-24-46963

Title: Relationships between hemoglobin levels at admission and adverse maternal and perinatal outcomes in preeclampsia patients: A retrospective study

Journal: PLOS ONE

Corresponding Author: Tongqiang He, E-mail: xbfymicu@163.com

Dear Dr. Ali Cetin,

We sincerely appreciate the reviewer’s constructive comments and the opportunity to revise our manuscript. We have revised the manuscript and answered reviewer's questions/comments with point by point hereunder according to the requirements in your letter.

To easily distinguish my answers from reviewer’s questions/comments, we highlighted all our answers in blue while keeping your letter and reviewer’s questions/comments in black in the Response letter.

Thank you again for your time and consideration.

Looking forward to hearing from you.

Best regards,

Tongqiang He

Department of Obstetrics and Gynecology Intensive Care Unit, Northwest Women's and Children's Hospital

Email: xbfymicu@163.com

Journal Requirements:

Reply: Thank you very much for your patient guidance. We have carefully checked the revised manuscript again to ensure that the format and file meet PLOS ONE's style requirements.

Reply: We apologize for the discrepancy between the ‘Funding Information’ and ‘Financial Disclosure’ sections. The funding information is hereby confirmed as follows: This project is supported by the Key Research and Development Program of Shaanxi China (Program number 2024SF-YBXM-230). The funders have no role in study design, data collection and analysis, the decision to publish, or the preparation of the manuscript. No author receives a salary from any of the funders. The Key Research and Development Program of Shaanxi is a sub-organization of Shaanxi Provincial Science and Technology Department. This inconsistency may be due to an oversight on our part when submitting the manuscript. We have corrected our funding information in the cover letter.

3. Please note that funding information should not appear in the Acknowledgments section or other areas of your manuscript. We will only publish funding information present in the Funding Statement section of the online submission form. Please remove any funding-related text from the manuscript.

Reply: Thank you for reminding us. We have removed the relevant financial information from my manuscript.

4. We note that your Data Availability Statement is currently as follows:

“All relevant data are within the manuscript and its Supporting Information files.”

Reply: Due to privacy and ethical restrictions, data cannot be shared publicly. However, researchers meeting the criteria for access to relevant data may obtain it from the Ethics Committee of Northwest Women's and Children's Hospital via email at xbfyyxll@163.com.

Reply: I'm sorry, but due to restrictions imposed by the ethics committee of our hospital, and the fact that our current research involves this raw data. The raw data cannot be shared publicly. We will update our data availability statement in the revised manuscript.

6. Please include a separate caption for figure 1 in your manuscript.

Reply: We have added a separate caption for Figure 1 in the revised manuscript.

Reply: The reviewers did not request to cite specific published works.

Additional Editor Comments:

I have reviewed this retrospective study examining hemoglobin levels and adverse outcomes in preeclampsia patients, along with two peer reviewer reports. One reviewer recommends rejection with strong methodological concerns, while the other suggests minor revision with more optimistic assessment. After careful consideration, I find this manuscript addresses a clinically relevant question with adequate sample size but has significant limitations preventing publication in its current form.

Reply: We are very grateful for your valuable constructive comments on our manuscript and thank you for giving us the opportunity to revise it. We are deeply appreciative and have made detailed revisions based on your review comments.

The most critical deficiency is exclusive use of univariate analysis. Contemporary medical research requires multivariable adjustment for observational studies examining exposure-outcome associations. The absence of logistic regression controlling for confounders such as maternal age, BMI, gestational age, and preeclampsia severity renders all reported odds ratios unreliable. Without these adjustments, conclusions cannot be considered valid.

Reply: The limitation of our study is that we only performed a univariate binary logistic regression analysis. We appreciate you pointing out this limitation and giving us the opportunity to revise it. We reanalyzed some of the data, such as categorizing the number of previous deliveries into two groups based on nulliparous and multiparous. Additionally, we classified preeclampsia severity as either mild or severe. These results are presented in Table 1. To minimize the influence of confounding factors on the impact of anemia and high hemoglobin level on adverse pregnancy outcomes in preeclampsia patients, we included maternal age, pre-pregnancy BMI, gestational age at onset of preeclampsia and the presence of severe preeclampsia, nulliparas, and assisted reproduction as covariates in our binary logistic regression analysis. The results of the multivariate regression analysis indicate that specific adverse pregnancy outcomes are associated with hemoglobin levels. Compared with preeclampsia patients with normal hemoglobin levels, patients with anemia at admission were at a greater risk for postpartum hemorrhage (OR: 3.800; 95% CI: 1.677–8.610) and cardiac dysfunction (OR: 2.860; 95% CI: 0.979–8.356). Moreover, patients with high hemoglobin levels at admission were at an increased risk for developing HELLP syndrome (OR: 2.503; 95% CI: 1.198–5.229). With respect to perinatal outcomes, we found that preeclampsia patients with high hemoglobin levels were at an increased risk of SGA and NICU admission, whereas patients with anemia were at a reduced risk. In addition, preeclampsia patients with high hemoglobin levels were at an increased risk of neonatal asphyxia (OR: 2.046; 95% CI: 1.107–3.784). To present the research results more intuitively, we used a forest plot to visually display the OR (95% CI) of the association between anemia and high hemoglobin levels with adverse pregnancy outcomes in Fig 2.

This manuscript exhibits concerning similarity to previously published literature, particularly in introduction and discussion sections. Multiple passages use language closely mirroring existing publications, suggesting inadequate paraphrasing or potential plagiarism. Most journals require similarity indices below 15%. Additionally, several sections display characteristics consistent with AI-generated writing, including unnaturally polished prose and formulaic sentence structures that lack authentic scientific voice.

Reply: When we first wrote this paper, we borrowed sentence structures from related articles and used basic translation software. Although we carefully revised the paper, some sentences still sound awkward because English is not our native language. We reduced the repetition rate in the revised manuscript. To improve its quality, we contacted a professional English editing company to make the English more accurate and correct basic errors.

The single-center retrospective design from a tertiary referral hospital introduces substantial selection bias. While authors acknowledge receiving "critically ill pregnant women," they inadequately address how this population differs from typical preeclampsia patients, limiting generalizability. Multiple comparison testing without statistical correction inflates false-positive risk, and outcome definitions like "early-stage heart failure" lack specificity.

Reply: In this study, 90.6% of pregnant women developed severe preeclampsia, a proportion that exceeds the 65.5% rate of severe preeclampsia observed among preeclampsia patients in China(1). We analyzed severe preeclampsia, maternal age, pre-pregnancy BMI, gestational age at onset of preeclampsia and the presence of nulliparas, and assisted reproduction as covariates in the multivariate binary logistic regression analysis, which to some extent reduced the impact of the severity of preeclampsia on the results. Despite the potential for selection bias, the study's substantial sample size offers a valuable opportunity to address some clinical concerns with hemoglobin levels, particularly in tertiary hospitals with a high proportion of patients experiencing severe preeclampsia. Future multicenter studies with larger samples are necessary to further investigate the impact of hemoglobin levels on adverse pregnancy outcomes in patients with preeclampsia.

In Chinese obstetrics and gynecology textbooks, early heart failure is a clinical concept with diagnostic criteria that include: 1. Chest tightness, palpitations, and shortness of breath after minimal activity.2. Resting heart rate greater than 110 beats per minute and respiratory rate greater than 20 breaths per minute.3. Frequent need to sit up to breathe or go to a window for fresh air at night due to chest tightness.4. Persistent fine crackles in the lungs that do not disappear after coughing. The diagnosis of early heart failure can be confusing to obstetricians outside China, so we have combined early heart failure and heart failure into cardiac dysfunction.

Data presentation has notable deficiencies including missing confidence intervals in Table 2, inadequate statistical methods description, and no power calculation. The temporal relationship between hemoglobin measurement and adverse outcomes is unclear, raising causation questions.

Reply: The incidence of adverse pregnancy outcomes in preeclampsia patients with different hemoglobin levels has been meticulously delineated in Table 2. In the revised manuscript, we have incorporated the 95% confidence intervals for the incidence rates of various adverse pregnancy outcomes, chi-square values, and post-hoc power analyses. However, due to the low incidence rates or minimal intergroup differences in some adverse pregnancy outcomes, the post-hoc power was less than 80%. However, we should also note the limitations of post-hoc power. Insignificant results mathematically guarantee low post-hoc power, which does not provide any ad

---

## [Editor Report · Decision Letter 1]

26 Sep 2025

PONE-D-24-46963R1Relationships between hemoglobin levels at admission and adverse maternal and perinatal outcomes in preeclampsia patients: A retrospective studyPLOS ONE

Dear Dr. He,

Thank you for submitting your manuscript to PLOS ONE. After careful consideration, we feel that it has merit but does not fully meet PLOS ONE’s publication criteria as it currently stands. Therefore, we invite you to submit a revised version of the manuscript that addresses the points raised during the review process.

We look forward to receiving your revised manuscript.

Kind regards,

Ali Cetin

Academic Editor

PLOS ONE

Journal Requirements:

Additional Editor Comments:

In the present form, the manuscript is considerably improved but not finalized. I would like to give you a warning, do not think that it is enough for your article to be corrected by Editing Firms, after all, these people are mostly not obstetricians, in obstetrics, a correct intervention or a correct stitch has a life-saving importance.

Finalize your article by examining the following details related to your article.

# Comprehensive Manuscript Corrections

## "Relationships between hemoglobin levels at admission and adverse maternal and perinatal outcomes in preeclampsia patients"

---

## **1. CRITICAL CORRECTIONS - Missing Decimal Points**

### Abstract Section:

- **Line with SGA**: "SGA(OR: 731; 95% CI: 0.517--1.032)" → **"SGA (OR: 0.731; 95% CI: 0.517-1.032)"**

- **Line with NICU**: "NICU admission(OR: 737; 95% CI: 0.567--0.960)" → **"NICU admission (OR: 0.737; 95% CI: 0.567-0.960)"**

---

## **2. GROUP NAMING STANDARDIZATION**

### Throughout the manuscript, replace:

- "anemia group" → **"Low Hb group"**

- "normal hemoglobin level group" → **"Normal Hb group"**

- "high hemoglobin level group" → **"High Hb group"**

### Alternative (if preferred):

- **Group A (anemia)** - hemoglobin ≤109 g/L

- **Group N (normal)** - hemoglobin 110-129 g/L

- **Group H (high)** - hemoglobin ≥130 g/L

---

## **3. SPACING CORRECTIONS - Add Space Before All Parentheses**

### Abstract:

- "infants(1--3)" → **"infants (1-3)"**

- "China(4)" → **"China (4)"**

- "infections(5,6)" → **"infections (5, 6)"**

- "age (SGA)(7--9)" → **"age (SGA) (7-9)"**

- "pregnancies(10,11)" → **"pregnancies (10, 11)"**

- "outcomes(12--14)" → **"outcomes (12-14)"**

- "changes(15)" → **"changes (15)"**

### Methods:

- "tests(creatinine ≥90 µmol/L)" → **"tests (creatinine ≥90 µmol/L)"**

- "dysfunction(NICU)" → **"dysfunction (NICU)"**

- "SGA(defined as" → **"SGA (defined as"**

- "delivery(defined as" → **"delivery (defined as"**

- "delivery(delivery before" → **"delivery (delivery before"**

- "unit(NICU)" → **"unit (NICU)"**

### Discussion:

- "dysfunction(20)" → **"dysfunction (20)"**

- "flow(19)" → **"flow (19)"**

- "anemia(21--23)" → **"anemia (21-23)"**

- "dysfunction(24--26)" → **"dysfunction (24-26)"**

- "dysfunction(27)" → **"dysfunction (27)"**

- "anemia(28)" → **"anemia (28)"**

- "fraction(29)" → **"fraction (29)"**

- All other reference citations need space before parenthesis

---

## **4. ETHICS APPROVAL SECTION REVISION**

### Current (problematic):

"Our study complied with the Declaration of Helsinki and was approved by the Ethics Committee of Northwest Women's and Children's Hospital. The ethics committee approved the waiving of informed consent for this research, as it was a retrospective study. The approval number was 2024-012. Patient information was anonymous during the analysis process."

### **Corrected version:**

"This study complied with the Declaration of Helsinki and was approved by the Ethics Committee of Northwest Women's and Children's Hospital (approval number: 2024-012). Informed consent was waived by the ethics committee due to the retrospective nature of the study. All patient information was anonymized during the analysis."

---

## **5. TABLE 1 CORRECTIONS**

1. "Severe preeclampsia, n(%)" → **"Severe preeclampsia, n (%)"**

2. "Nulliparas, n(%)" → **"Nulliparas, n (%)"**

3. "Assisted reproduction, n(%)" → **"Assisted reproduction, n (%)"**

4. All values: "31.17±4.63" → **"31.17 ± 4.63"** (add spaces around ±)

5. Statistics: "*F*=*0.419*" → **"F = 0.419"** (remove italics from equals sign)

6. "*χ^2^*=*6.744*" → **"χ² = 6.744"** (proper chi-square symbol)

### Column Headers:

- "Anemia group" → **"Low Hb group (n=395)"**

- "Normal hemoglobin level group" → **"Normal Hb group (n=877)"**

- "High hemoglobin level group" → **"High Hb group (n=443)"**

---

## **6. TABLE 2 CORRECTIONS**

1. "Rate,%(95% CI)" → **"Rate, % (95% CI)"**

2. All CI values need spaces: "2.0(0.9,4.0)" → **"2.0 (0.9, 4.0)"**

3. "Post hoc power,%" → **"Post hoc power, %"**

4. Apply same column header changes as Table 1

---

## **7. STATISTICAL NOTATION STANDARDIZATION**

Throughout manuscript:

- "P value<0.05" → **"P < 0.05"**

- "P values>0.05" → **"P > 0.05"**

- "(OR:3.800; 95% CI:1.677--8.610)" → **"(OR: 3.800; 95% CI: 1.677-8.610)"**

- "(OR:2.224; 95% CI:0.910--5.432)" → **"(OR: 2.224; 95% CI: 0.910-5.432)"**

- "(OR:2.451; 95% CI:1.140--5.268)" → **"(OR: 2.451; 95% CI: 1.140-5.268)"**

---

## **8. RESULTS SECTION CORRECTIONS**

### Paragraph 1:

- "120.27 g/L(SD 15.52)" → **"120.27 g/L (SD 15.52)"**

- "33.98 weeks(SD 4.18)" → **"33.98 weeks (SD 4.18)"**

### Paragraph 2:

- "The study population included 395 patients with anemia, 877 patients with normal hemoglobin levels, and 443 patients with high hemoglobin levels."

→ **"The study population comprised 395 patients in the Low Hb group, 877 in the Normal Hb group, and 443 in the High Hb group."**

---

## **9. DISCUSSION SECTION STYLE IMPROVEMENTS**

1. "A series of studies have demonstrated" → **"Several studies have demonstrated"**

2. "Interestingly, our study revealed" → **"Notably, our study revealed"**

3. "These findings contradict the results" → **"These findings contrast with the results"**

4. "Owing to" → **"Due to"** (throughout)

5. "data concerning" → **"data regarding"**

6. "comparatively little research has been devoted" → **"relatively little research has been conducted"**

---

## **10. REFERENCES FORMATTING**

### Standardize all references:

1. Month names: Choose either abbreviated (Jun, Sept, Nov) or full (June, September, November) - be consistent

2. Page ranges: Use en-dash (–) not hyphen (-): "47--68" → **"47–68"**

3. Add spaces: "2019 Aug;1450(1):47--68" → **"2019 Aug; 1450(1): 47–68"**

4. Ensure consistent punctuation after author initials

---

## **11. TERMINOLOGY CONSISTENCY**

Choose one and use throughout:

- "preeclampsia patients" OR **"patients with preeclampsia"** (preferred)

- "hemoglobin concentration" OR **"hemoglobin level"** (choose one)

- "pre-pregnancy BMI" OR **"prepregnancy BMI"** (choose one)

- "post-partum" → **"postpartum"** (no hyphen)

---

## **12. NUMBER FORMATTING**

- Range dashes: "12--14" → **"12-14"** (use en-dash consistently)

- "3%--5%" → **"3%-5%"**

- Ensure leading zeros: ".731" → **"0.731"**

---

## **13. ABBREVIATION ISSUES TO FIX**

1. **ANOVA** - define at first use

2. Remove redundant definitions of:

- BMI (defined multiple times)

- NICU (defined multiple times)

- SGA (defined multiple times)

- HELLP syndrome (defined multiple times)

---

## **14. MINOR BUT IMPORTANT CORRECTIONS**

1. Methods section: Add period after "control" in Abstract Methods

2. "24-hour urine protein" → **"24-h urine protein"** (for consistency)

3. ">40 IU/L" → **"> 40 IU/L"** (add space after >)

4. "≥90 µmol/L" → **"≥ 90 µmol/L"** (add space after ≥)

---

## **SUMMARY OF KEY CHANGES**

1. **Add spaces** before ALL parentheses

2. **Fix missing decimal points** (0.731, 0.737)

3. **Standardize group names** (Low Hb, Normal Hb, High Hb)

4. **Revise Ethics section** to eliminate redundancy

5. **Fix table formatting** (spaces, statistical notation)

6. **Standardize statistical reporting** (OR: X.XXX; 95% CI: X.XXX-X.XXX)

7. **Choose consistent terminology** throughout

8. **Fix reference formatting** for journal requirements

9. **Under each table, write the definitions of all abbreviations in that table separately.

These corrections will ensure the manuscript meets Q1 journal publication standards.

---

## [Author Response · Author response to Decision Letter 2]

3 Oct 2025

PONE-D-24-46963R1

Relationships between hemoglobin levels at admission and adverse maternal and perinatal outcomes in preeclampsia patients: A retrospective study

PLOS ONE

Dear Dr. He,

Thank you for submitting your manuscript to PLOS ONE. After careful consideration, we feel that it has merit but does not fully meet PLOS ONE’s publication criteria as it currently stands. Therefore, we invite you to submit a revised version of the manuscript that addresses the points raised during the review process.

We look forward to receiving your revised manuscript.

Kind regards,

Ali Cetin

Academic Editor

PLOS ONE

Response to Reviewers’ Comments

Manuscript ID: PONE-D-24-46963R1

Title: Relationships between hemoglobin levels at admission and adverse maternal and perinatal outcomes in patients with preeclampsia

Journal: PLOS ONE

Corresponding Author: Tongqiang He, E-mail: xbfymicu@163.com

Dear Dr. Ali Cetin,

We greatly appreciate your affirmation of our work revising the manuscript, as well as the detailed suggestions you provided for its improvement. According to the requirements in your letter, we have revised the manuscript point by point, as detailed below.

To easily distinguish my answers from editor's comments, we highlighted all our answers in blue while keeping the comments of your letter in black in the Response letter.

Thank you again for your time and consideration.

Looking forward to hearing from you.

Best regards,

Tongqiang He

Department of Obstetrics and Gynecology Intensive Care Unit, Northwest Women's and Children's Hospital

Email: xbfymicu@163.com

Journal Requirements:

Reply: The reviewers did not request the citation of specific previously published works.

Reply: We re-retrieved the references cited in the manuscript via PubMed and removed citations of two articles due to incomplete bibliographic information. The relevant details were indicated in the revised manuscript with track changes.

Additional Editor Comments:

In the present form, the manuscript is considerably improved but not finalized. I would like to give you a warning, do not think that it is enough for your article to be corrected by Editing Firms, after all, these people are mostly not obstetricians, in obstetrics, a correct intervention or a correct stitch has a life-saving importance.

Reply: Thank you very much for your reminder. We fully recognize that professional English editing services can help non-native English-speaking researchers clearly convey their research ideas, but they are not the deciding factor in whether a manuscript is published. The scientific content of the research is the main reason for acceptance. Thank you again for your detailed guidance on our manuscript. Following your suggestions, we have carefully revised the manuscript.

Finalize your article by examining the following details related to your article.

# Comprehensive Manuscript Corrections

## "Relationships between hemoglobin levels at admission and adverse maternal and perinatal outcomes in preeclampsia patients"

Reply: We have chosen to use "patients with preeclampsia" throughout the manuscript. Therefore, we revised the manuscript's title to "Relationships between hemoglobin levels at admission and adverse maternal and perinatal outcomes in patients with preeclampsia".

---

## **1. CRITICAL CORRECTIONS - Missing Decimal Points**

### Abstract Section:

- **Line with SGA**: "SGA(OR: 731; 95% CI: 0.517--1.032)" → **"SGA (OR: 0.731; 95% CI: 0.517-1.032)"**

- **Line with NICU**: "NICU admission(OR: 737; 95% CI: 0.567--0.960)" → **"NICU admission (OR: 0.737; 95% CI: 0.567-0.960)"**

Reply: First, we would like to sincerely apologize for the oversight. We have corrected the two instances of missing decimal points in the abstract section of the manuscript.

---

## **2. GROUP NAMING STANDARDIZATION**

### Throughout the manuscript, replace:

- "anemia group" → **"Low Hb group"**

- "normal hemoglobin level group" → **"Normal Hb group"**

- "high hemoglobin level group" → **"High Hb group"**

### Alternative (if preferred):

- **Group A (anemia)** - hemoglobin ≤109 g/L

- **Group N (normal)** - hemoglobin 110-129 g/L

- **Group H (high)** - hemoglobin ≥130 g/L

Reply: Throughout the manuscript, the terms "anemia group," "normal hemoglobin level group," and "high hemoglobin level group" were replaced with "Low Hb group", "Normal Hb group", and "High Hb group".

---

## **3. SPACING CORRECTIONS - Add Space Before All Parentheses**

### Abstract:

- "infants(1--3)" → **"infants (1-3)"**

- "China(4)" → **"China (4)"**

- "infections(5,6)" → **"infections (5, 6)"**

- "age (SGA)(7--9)" → **"age (SGA) (7-9)"**

- "pregnancies(10,11)" → **"pregnancies (10, 11)"**

- "outcomes(12--14)" → **"outcomes (12-14)"**

- "changes(15)" → **"changes (15)"**

### Methods:

- "tests(creatinine ≥90 µmol/L)" → **"tests (creatinine ≥90 µmol/L)"**

- "dysfunction(NICU)" → **"dysfunction (NICU)"**

- "SGA(defined as" → **"SGA (defined as"**

- "delivery(defined as" → **"delivery (defined as"**

- "delivery(delivery before" → **"delivery (delivery before"**

- "unit(NICU)" → **"unit (NICU)"**

### Discussion:

- "dysfunction(20)" → **"dysfunction (20)"**

- "flow(19)" → **"flow (19)"**

- "anemia(21--23)" → **"anemia (21-23)"**

- "dysfunction(24--26)" → **"dysfunction (24-26)"**

- "dysfunction(27)" → **"dysfunction (27)"**

- "anemia(28)" → **"anemia (28)"**

- "fraction(29)" → **"fraction (29)"**

- All other reference citations need space before parenthesis

Reply: Thank you for your thorough and patient guidance. We have added space before all parentheses in the manuscript.

---

## **4. ETHICS APPROVAL SECTION REVISION**

### Current (problematic):

"Our study complied with the Declaration of Helsinki and was approved by the Ethics Committee of Northwest Women's and Children's Hospital. The ethics committee approved the waiving of informed consent for this research, as it was a retrospective study. The approval number was 2024-012. Patient information was anonymous during the analysis process."

### **Corrected version:**

"This study complied with the Declaration of Helsinki and was approved by the Ethics Committee of Northwest Women's and Children's Hospital (approval number: 2024-012). Informed consent was waived by the ethics committee due to the retrospective nature of the study. All patient information was anonymized during the analysis."

Reply: We have updated the ethics approval to the correct version in the manuscript.

---

## **5. TABLE 1 CORRECTIONS**

1. "Severe preeclampsia, n(%)" → **"Severe preeclampsia, n (%)"**

2. "Nulliparas, n(%)" → **"Nulliparas, n (%)"**

3. "Assisted reproduction, n(%)" → **"Assisted reproduction, n (%)"**

Reply: We have added space before all parentheses in table 1.

4. All values: "31.17±4.63" → **"31.17 ± 4.63"** (add spaces around ±)

Reply: We have added spaces around all "±" in table 1.

5. Statistics: "*F*=*0.419*" → **"F = 0.419"** (remove italics from equals sign)

Reply: We have removed italics from equal signs and added spaces around equal signs in Table 1. Only the italicized format of "F" has been retained.

6. "*χ^2^*=*6.744*" → **"χ² = 6.744"** (proper chi-square symbol)

Reply: We have corrected the format of the chi-square symbol and added spaces around equal signs.

### Column Headers:

- "Anemia group" → **"Low Hb group (n=395)"**

- "Normal hemoglobin level group" → **"Normal Hb group (n=877)"**

- "High hemoglobin level group" → **"High Hb group (n=443)"**

Reply: We have revised the column headers of table 1 according to your comments.

---

## **6. TABLE 2 CORRECTIONS**

1. "Rate,%(95% CI)" → **"Rate, % (95% CI)"**

2. All CI values need spaces: "2.0(0.9,4.0)" → **"2.0 (0.9, 4.0)"**

3. "Post hoc power,%" → **"Post hoc power, %"**

Reply: Thanks for your careful guidance. We have added spaces in the appropriate locations as per your comments.

4. Apply same column header changes as Table 1

Reply: We have modified the column headers in Table 2 as requested.

---

## **7. STATISTICAL NOTATION STANDARDIZATION**

Throughout manuscript:

- "P value<0.05" → **"P < 0.05"**

- "P values>0.05" → **"P > 0.05"**

Reply: All P values in the manuscript have been modified to P.

- "(OR:3.800; 95% CI:1.677--8.610)" → **"(OR: 3.800; 95% CI: 1.677-8.610)"**

- "(OR:2.224; 95% CI:0.910--5.432)" → **"(OR: 2.224; 95% CI: 0.910-5.432)"**

- "(OR:2.451; 95% CI:1.140--5.268)" → **"(OR: 2.451; 95% CI: 1.140-5.268)"**

Reply: We have made the requested modifications throughout the manuscript.

---

## **8. RESULTS SECTION CORRECTIONS**

### Paragraph 1:

- "120.27 g/L(SD 15.52)" → **"120.27 g/L (SD 15.52)"**

- "33.98 weeks(SD 4.18)" → **"33.98 weeks (SD 4.18)"**

Reply: We have added space before parentheses in paragraph 1 of the results section.

### Paragraph 2:

- "The study population included 395 patients with anemia, 877 patients with normal hemoglobin levels, and 443 patients with high hemoglobin levels."

→ **"The study population comprised 395 patients in the Low Hb group, 877 in the Normal Hb group, and 443 in the High Hb group."**

Reply: We have made the requested modification in paragraph 2 of the results section.

---

## **9. DISCUSSION SECTION STYLE IMPROVEMENTS**

1. "A series of studies have demonstrated" → **"Several studies have demonstrated"**

2. "Interestingly, our study revealed" → **"Notably, our study revealed"**

3. "These findings contradict the results" → **"These findings contrast with the results"**

4. "Owing to" → **"Due to"** (throughout)

5. "data concerning" → **"data regarding"**

6. "comparatively little research has been devoted" → **"relatively little research has been conducted"**

Reply: Thanks for your guidance. We have made the requested modifications throughout the manuscript.

---

## **10. REFERENCES FORMATTING**

### Standardize all references:

1. Month names: Choose either abbreviated (Jun, Sept, Nov) or full (June, September, November) - be consistent

Reply: We uniformly adopt the abbreviated format for months in the reference section.

2. Page ranges: Use en-dash (–) not hyphen (-): "47--68" → **"47–68"**

3. Add spaces: "2019 Aug;1450(1):47--68" → **"2019 Aug; 1450(1): 47–68"**

4. Ensure consistent punctuation after author initials

Reply: We have carefully reviewed the reference formatting and corrected some errors as per your comments.

---

## **11. TERMINOLOGY CONSISTENCY**

Choose one and use throughout:

- "preeclampsia patients" OR **"patients with preeclampsia"** (preferred)

Reply: We have chosen to use "patients with preeclampsia" throughout the manuscript.

- "hemoglobin concentration" OR **"hemoglobin level"** (choose one)

Reply: We have chosen to use "hemoglobin level" throughout the manuscript.

- "pre-pregnancy BMI" OR **"prepregnancy BMI"** (choose one)

Reply: We have chosen to use "pre-pregnancy BMI" throughout the manuscript.

- "post-partum" → **"postpartum"** (no hyphen)

Reply: We removed the hyphen in "post-partum".

---

## **12. NUMBER FORMATTING**

- Range dashes: "12--14" → **"12-14"** (use en-dash consistently)

- "3%--5%" → **"3%-5%"**

Reply: We have replaced all hyphens indicating ranges with en-dashes throughout the manuscript.

- Ensure leading zeros: ".731" → **"0.731"**

Reply: We have corrected the two instances of missing zeros and decimal points in the abstract section of the manuscript.

---

## **13. ABBREVIATION ISSUES TO FIX**

1. **ANOVA** - define at first use

Reply: Since ANOVA appeared only once in the statistical methods section, we removed the abbreviation of ANOVA and retained only the full term “analysis of variance”.

2. Remove redundant definitions of:

- BMI (defined multiple times)

- NICU (defined multiple times)

- SGA (defined multiple times)

- HELLP syndrome (defined multiple times)

Reply: We carefully examined the manuscript and found that these redundant definitions of abbreviations likely occurred because we initially placed table footnotes within the main text. During revisions, we relocated the footnote content to the tables themselves.

---

## **14. MINOR BUT IMPORTANT CORRECTIONS**

1. Methods section: Add period after "control" in Abstract Methods

Reply: We added a period after "control" in Abstract Methods.

2. "24-hour urine protein" → **"24-h urine protein"** (for consistency)

3. ">40 IU/L" → **"> 40 IU/L"** (add space after >)

4. "≥90 µmol/L" → **"≥ 90 µmol/L"** (add space after ≥)

Reply: We have made the requested modifications in the manuscript.

---

## **SUMMARY OF KEY CHANGES**

1. **Add spaces** before ALL parentheses

2. **Fix missing decimal points** (0.731, 0.737)

3. **Standardize group names** (Low Hb, Normal Hb, High Hb)

4. **Revise Ethics section** to eliminate redundancy

5. **Fix table formatting** (spaces, statistical notation)

6. **Standardize statistical reporting** (OR: X.XXX; 95% CI: X.XXX-X.XXX)

7. **Choose consistent terminology** throughout

8. **Fix reference formatting** for journal requirements

9. **Under each table, write the definitions of all abbreviations in that table separately.

These corrections will ensure the manuscript meets Q1 journal publication standards.

Reply: Thank you once again for your detailed guidance, which has been invaluable in enhancing our manuscript. We have made the requested revisions in full accordance with your instructions.

While revising your submission, please upload your figure files to the Preflight Analysis and Conversion Engine (PACE) digital diagnostic tool, https://pac

---

## [Editor Report · Decision Letter 2]

6 Oct 2025

Relationships between hemoglobin levels at admission and adverse maternal and perinatal outcomes in patients with preeclampsia

PONE-D-24-46963R2

Dear Dr. He,

We’re pleased to inform you that your manuscript has been judged scientifically suitable for publication and will be formally accepted for publication once it meets all outstanding technical requirements.

Kind regards,

Ali Cetin

Academic Editor

PLOS ONE

Additional Editor Comments (optional):

I reread your revisions. I found them adequate.
---

## [Editor Report · Acceptance letter]

PONE-D-24-46963R2

PLOS ONE

Dear Dr. He,

I'm pleased to inform you that your manuscript has been deemed suitable for publication in PLOS ONE. Congratulations! Your manuscript is now being handed over to our production team.

Kind regards,

on behalf of

Professor Ali Cetin

Academic Editor

PLOS ONE